# End-of-Life Care Training for Patients with Traumatic Brain Injury in Ghana: A Novel Curriculum and Its Initial Implementation

**DOI:** 10.3390/jcm14113643

**Published:** 2025-05-22

**Authors:** John Bruno, Mayur Patel, Rebecca Henderson, Michael Mathelier, Taylor N. Smith, Joseph C. Pompa, Cassandra Clay, Marie-Carmelle Elie, Sheba Afi Mansa Fiadzomor, Lawrence Nsohlebna Nsoh, Torben K. Becker

**Affiliations:** 1Division of Critical Care Medicine, Department of Emergency Medicine, College of Medicine, University of Florida, 1600 Archer Road, Gainesville, FL 32608, USA; 2Section of Global Health, Department of Emergency Medicine, College of Medicine, University of Florida, 1600 Archer Road, Gainesville, FL 32608, USA; 3Department of Neurology, University of Florida, 1600 Archer Road, Gainesville, FL 32608, USA; rrhenderson@ufl.edu; 4Department of Obstetrics & Gynecology, University of Alabama, Tuscaloosa, AL 35487, USA; 5College of Medicine, University of Florida, 1600 Archer Road, Gainesville, FL 32608, USA; 6College of Nursing, University of Florida, 1600 Archer Road, Gainesville, FL 32608, USA; 7Department of Emergency Medicine, Saint Francis Hospital, Roslyn, NY 11576, USA; 8Department of Emergency Medicine, University of Alabama at Birmingham, Birmingham, AL 35294, USA; dr.s.fiadzomor@gmail.com (S.A.M.F.); nsohabolga@gmail.com (L.N.N.); 9Ghana Armed Forces Medical Services, Ghana

**Keywords:** palliative care, low- and middle-income countries, end-of-life, traumatic brain injury

## Abstract

The implementation and practice of palliative medicine have numerous boundaries in low- and middle-income countries (LMICs), stemming from various cultural, legal, and religious concerns. Additionally, professional education in palliative care medicine in these countries is severely lacking, especially when compared with developed countries. **Background/Objectives**: To enhance and demystify palliative medicine practice to health care providers in LMICs. **Methods**: We developed a novel and comprehensive course in palliative care medicine and end-of-life (EOL) care, specifically within the context of management of patients with traumatic brain injury (TBI). We performed both immediate pre-course and post-course analysis of course participant comprehension and feedback, as well as a one-year post-course analysis and small group discussion. **Results**: The comprehension of the course material was strong, as participants scored an average of 13.9 points better on the post-test compared to the pre-test (49.6% vs. 35.7%, *p* < 0.001). Participants in the one-year follow-up session reported long-term applicability of the course material in their respective practice settings, with all participants reporting that they utilize the course material often. Small group discussion responses indicated a strong level of comprehension of the course material. **Conclusions**: Providing education in palliative medicine to health care professionals in LMICs is feasible, and likely to be both well-received and strongly influential to local medical practice. Local cultural and religious practices may be less of a barrier to the provision of palliative medicine than previously considered. Practicing palliative medicine, particularly at EOL, may strengthen patient–provider relationships, improve job satisfaction among health care providers, and improve the perception of medical care provided in LMIC medical settings.

## 1. Introduction

Morbidity and mortality are common after hospital admissions worldwide, especially considering an aging global population and the increasing prevalence of chronic end-stage conditions requiring high levels of care. This is especially true in low- and middle-income countries (LMICs), in particular sub-Saharan African countries, where in-hospital mortality can be as high as 40-fold than that seen in high-resource countries, owing to a high burden of severe disease in combination with limited availability of medical resources [1]. The provision of palliative care in conjunction with curative care has been demonstrated to reduce unnecessary treatments or procedures, reduce health care costs, and improve patient satisfaction without increasing overall mortality [2,3,4]. Evidence dating back more than 40 years supports that the provision of palliative care is beneficial not only to those with end-stage and life-limiting diseases as referenced above, but also to patients with other serious illnesses or injuries, both acute and chronic [5].

Most of the available literature on palliative medicine comes from North America, Europe, and Australia, with a vast majority of publications emerging from the United States. An urgent need for expanded access to palliative care has been identified in LMICs [5]. Similarly, while the practice of palliative medicine has seen significant advances in Western medicine, its utilization in LMIC medical practice has lagged behind substantially. The World Health Organization (WHO), which has identified palliative care as a basic human right, identified in 2011 over 34 million people who died from diseases requiring palliative medicine. This number is likely underestimated considering the breadth of diseases that may benefit from palliative care, as well as the underreporting of epidemiologic statistics [5,6]. Current estimates indicate that as high as 80% of the world’s palliative care needs come from LMICs, a number that may increase further as the world’s population ages [7]. However, its implementation, and consequently patient access to this highly beneficial care, are often highly limited. This is owing to local legal, cultural, and religious factors in addition to the paucity of literature arising from these regions [6,7,8,9,10,11,12]. Resource limitations also drive significant disparities regarding the availability of clinicians with dedicated end-of-life skills and knowledge, access to appropriate pain management solutions, and a lack of public policies and national health system support for patients at the end of their lives. Furthermore, formal or even informal education in palliative medicine is either lacking or non-existent in many LIMC teaching facilities [13,14,15,16]. Similarly, health care professionals in LMICs frequently view their knowledge and training in palliative concepts as notably poor [17,18,19,20].

The burden of critical illness due to trauma, including those with traumatic brain injury (TBI), is high in LMICs [21,22]. Ghana, a West African country with a population of approximately 31 million, is a stable democracy and is classified as a low- to middle-income country (LMIC). Many of the challenges in end-of-life care mentioned above limit patients’ access to palliative medicine in Ghana. The Ghana Armed Forces Medical Services provide medical and surgical care, including emergency and intensive care unit services, to members of the military and civilians at military hospitals throughout the country and in other countries as part of peacekeeping and related operations [23]. The incidence of TBI is high in Ghana, due to a high burden of motor vehicle crashes (MVCs) [21]. Given the high burden of palliative care needs in patients suffering from TBI, we identified this patient population as of particular interest and of high potential for successful utilization of palliative care practice and techniques in Ghana [24]. Next, we describe the implementation of a short, comprehensive training course in the principles and practice of palliative and end-of-life care for patients with TBI, presented to health care professionals of multiple disciplines employed at a single urban military medical center in Ghana. Finally, we report on both immediate and one-year feedback and impressions from the students who attended the program.

## 2. Methods

We developed the end-of-life (EOL) care in the TBI didactic course as a curriculum containing 12 lectures, interspersed with 4 small group sessions designed to reinforce the lecture material (Table 1). Lectures were written by the authors based on current literature and best practice guidelines. The material was adapted as best as possible toward practice in a low-resource setting, with the omission of some diagnostic and treatment modalities unavailable or not easily available in LMICs. The lectures were presented by primarily emergency medicine physicians, most formal subspecialty training in either global emergency medicine, emergency medical services, emergency ultrasound, hospice and palliative care, critical care medicine, and neurology critical care medicine, with several authors having formally trained in several of the above subspecialties. The didactic material was presented over two days of lecture, loosely divided so that day 1 was focused on principles of TBI and issues related to complications and prognosis, and day 2 was focused on palliative medicine and end-of-life related issues (Table 1). TBI lecture topics specifically were written with a particular focus on the identification and control of distressing symptoms. The third and final day of the course was dedicated to group sessions and simulation, designed to incorporate all of the palliative care principles learned in the didactic portion of the course, with a focus on EOL care communication topics such as breaking bad news, goals of care discussions, and difficult family interactions.

A pre-test was administered, and an identical exam was given at the course’s conclusion to assess the comprehension of the didactic materials. Similarly, a pre- and post-course survey was administered to assess satisfaction with and potential applicability of the course material. Finally, a one-year follow-up small group discussion and survey were held with the course director and volunteers from the pool of students who had taken the course, during which another survey was administered (Appendix D). In this session with a focus group format, course graduates discussed if and how the course had affected their practice. They were also given the opportunity to discuss specific cases or scenarios in which the course material was utilized.

## 3. Results

A total of 33 health care professionals took the EOL care course as a multidisciplinary course consisting mostly of physicians of multiple levels of training, including attending physicians, clinical psychologists, and nurses. A total of 32 participants’ data were available for the pre-test and pre-course survey, and data were available for 31 participants for the post-test and post-survey (Appendix A, Appendix B and Appendix C). One post-test was omitted from the final analysis because the identity of the test-taker could not be determined, and thus it could not be correlated with a corresponding pre-test. Exams were scored with a total of 100 points achievable. Participants scored an average of 13.9 points better on the post-test compared to the pre-test (49.6% vs. 35.7%, *p* < 0.001; Table 2).

In the pre-survey, participants selected their pre-course knowledge, comfort level, and exposure to palliative care practices on a scale of 1–5. On average, participants ranked their pre-course palliative care knowledge as 2.5/5 (SD 0.9), indicating between “below average” and “average” knowledge. They indicated the frequency at which they practiced palliative-related medicine between “once per month” and “once per week”. Average pre and post-test scores are available in Table 3.

In the post-survey, participants assessed the content of the course (Appendix B). Notably, confidence in the course content was assessed on a scale of 1–4, with 4 being the highest. Course participants rated their post-course knowledge of the TBI content as an average of 3.7 (SD 0.5), and their knowledge of the palliative care content as an average of 3.8 (SD 0.4). Participants ranked the effectiveness of the lecture material and the small group sessions, on a scale of 1–5 (with 5 being the highest), an average of 4.6 (SD 0.8) and 4.7 (SD 0.8), respectively. Despite the intentional difficulty of the course material, pre-test and post-test, participants ranked the difficulty of the course as an average of 3.0 (SD 0.3), indicating a desired amount of difficulty, with a score of 1/5 indicating a course that was too difficult, and 5/5 indicating material that was too simple. Please see Appendix B.

A total of eleven participants volunteered to participate in the one-year follow-up session (Table A1 in Appendix E). In the one-year survey, participants reported on average that their knowledge base regarding TBI had greatly improved since taking the course (4.63/5), but reported variability regarding how often this knowledge was utilized (3.22/5). Participants noted that many of them are infrequently in settings in which they can manage TBI patients, except for participants who work in the emergency department.

In contrast, participants noted that they utilize the course material on palliative care and end-of-life communication highly frequently, with all participants noting that their perception of palliative medicine has greatly changed (5/5) since taking the course. All participants noted that they use content and strategies learned from the course in breaking bad news often (5/5). Participants reported that they utilize concepts learned from the palliative medicine section of the course highly frequently (4.9/5), with 9/11 participants noting that they use concepts learned in this course daily, including each of the nurses interviewed (Table 4). However, only 3/11 participants in these discussions reported using shared decision-making principles as taught in the course. Additionally, important qualitative data were also collected during this encounter, and will be discussed below.

## 4. Discussion

The World Health Organization has declared access to palliative medicine as a human right [5,7,25]. It defines palliative care as “… A care approach that improves the quality of life of patients and their families who are facing problems associated with a life-threatening illness, through prevention and relief of suffering by means of early identification and impeccable assessment of pain and other problems, physical, psychosocial and spiritual” [25,26]. Importantly, early provision of palliative care, particularly in the setting of critical illness, does not negatively impact overall mortality [2,3]. Despite this, the delivery of palliative care education to health care professionals working in LMICs, and consequently improving local patient access to palliative services, have been challenging [5,6,7,8,13,19,27].

We present the results of the implementation of a three-day lecture series focused on palliative care medicine to a class of health care professionals employed at a large military hospital in Ghana. Specifically, the course material was prepared in the context of traumatic brain injury patients—a patient population that is very common in Ghana. While there is literature describing the practice of palliative medicine, formal training in palliative medicine is lacking in Ghana. Our course thus provided a novel educational experience to participating Ghanaian health care providers in EOL medicine, which has not been previously described in the literature [11,12,28,29]. While there is formalized palliative education in some LMICs, for instance, South Africa, to our knowledge, this is the first documented literature on the implementation of such a curriculum to students in an LMIC setting without previous training in palliative management [14,27]. Notably, the students’ reflections on the content both immediately after the course and at a one-year follow-up after having the opportunity to potentially practice the knowledge learned provide unique insights into the impact of practice that even such a short course can provide.

Response to the course was overwhelmingly positive among participants, as evidenced by the improvement in test scores after taking the course, the post-test survey, and most importantly in the one-year follow-up. Immediate post-course test and survey results suggest both an appreciation of the course material presented, as well as a perceived relevance toward their practice settings. Perhaps more importantly, the one-year follow-up survey and interview results suggested a deep and foundational understanding of the course material presented, as evidenced by the high levels of practical application reported by course participants, and their perception of resultant improvements in their day-to-day practice.

The positive response was most evident in the context of material relating to palliative medicine, breaking bad news, and end-of-life care. Participants, who initially reported utilizing palliative care practice between once monthly and once weekly in the pre-test survey, highly utilized the concepts after the course completion. All participants interviewed at the one-year follow-up reported using a model, such as the SPIKES model (Setting-Perception-Invitation-Knowledge-Emotion-Summarize) discussed in the course, to aid in breaking bad news [30]. Participants noted that while they had previously considered breaking bad news to be one of the worst and most frustrating parts of their job, they now approach such situations and difficult conversations favorably as they view themselves as having a certain level of expertise in the subject. Participants also perceived that utilizing the palliative care approach from the course seemed to improve family and patient satisfaction. This is consistent with previous literature that suggests incorporating palliative medicine into routine practice is associated with increased job satisfaction in health care professionals [6,20]. Course participants also note that families seem to respond more positively than expected to bad news when presented using the tools and strategies discussed in the course. Interestingly, participants also noted that when coworkers who did not take the course are tasked with breaking bad news to families or having goals of care conversations, participants are now more likely to notice flaws in these approaches, and more importantly, consequent negative effects on and reactions from the family members involved in these discussions.

Interestingly, the material regarding shared decision-making was not as well received as the other course material and was not utilized with any notable frequency, according to participants in the one-year follow-up. Participants pointed to several factors that influenced this pattern. Firstly, they note that families often request therapies that are not evidence-based, including home remedies, or therapies that are either unavailable or unlikely to benefit the patient. They report that family members commonly have negative reactions when these therapies are not offered after attempts at shared and evidence-based informed decision-making. This is consistent with the experiences of health care professionals at other medical centers in LMICs [6,31]. Secondly, participants note that the up-front fee-for-service payment model at Ghanaian hospitals may influence families to choose less expensive treatment modalities, even though these specific modalities may not provide a benefit or may even be harmful to patients at the end of life. For instance, a participant noted that when families ultimately decide to pursue withdrawal of life-sustaining therapy, if given a choice, family members may elect not to treat the patients’ pain at the end of life due to financial constraints. Notably, narcotic analgesia medication remains expensive and less available in LMICs [32]. It is possible that as palliative medicine provision evolves in LMICs, and access to palliative care medications improves, shared decision-making may develop value in patient care. However, the role of shared decision-making would seem to have several important factors limiting its utility.

Initially, participants expressed skepticism during the course’s small group discussions that local cultural and religious practices may limit the ability to apply communication principles primarily derived in high-resource settings, specifically regarding topics involving death and morbidity. Indeed, local cultural and religious concerns have been previously cited as potential barriers to practicing palliative medicine [7,10,20,33]. However, these concerns may be less important than previously considered. Based on our findings, except for the content on shared decision-making, these considerations appear not to have significantly factored into the participants’ ability to impact patient care using palliative care principles. Participants noted that the palliative approach to breaking bad news and end-of-life care frequently triggered positive involvement in family religious and cultural practice, often helping family members come to terms with a bad outcome, and finding solace in their religion as a foundation to cope. Based on our findings, breaking bad news in a regimented fashion, and utilizing a palliative care approach toward end-of-life issues as taught in this course may ultimately help to positively engage the cultural and religious beliefs and practices of families. However, this area of inquiry requires more research before attempting to generalize these results, especially in regard to the legality of such practice locally, and the potential reactions of local communities and government bodies to the practice of palliative medicine [20,33].

### Limitations

There are several limitations to this study. The primary intent of this project was to report on the feasibility of a dedicated palliative care course, and specifically its effect on the practice patterns of Ghanaian health care providers. We did not intend this to be a clinical study, and for this reason, we did not investigate patient-centered outcomes such as effects of our course on mortality, lifespan, health care costs, or patient satisfaction at EOL. These data would be highly useful, and further studies on the topic would be beneficial to understanding the long-term effects on patients from the implementation of palliative medicine practice at Ghanaian health care facilities. Previous data available suggest that these patient-centered outcomes would have a high likelihood of being improved as a consequence of the implementation of our course [2,3,34].

Regarding the course itself, the level of difficulty (oriented towards physicians) of content material, including pre-/post-test material, may have led to lower comprehension, and therefore less clinical utilization, of concepts by participants with less rigorous education, such as nurses. However, post-test scores still indicated comprehension in these individuals, and post-course survey data suggested that the course was at a desired level of complexity. Notably, feedback provided by non-physician participants in the one-year follow-up group discussion indicated high levels of comprehension of the source material, suggesting that this material was certainly suitable for presentation to a non-physician audience. The creation of courses specifically tailored to different health care professions may improve comprehension. However, we observed that a multidisciplinary course led to highly valuable and meaningful small-group discussions and simulation experiences and may have enhanced the reflection of the course material.

While Ghana is home to many different languages and dialects, many Ghanaians speak English, and course participants were proficient in speaking English. Language barriers have been identified as a barrier to the practice of palliative medicine, with notable concern that palliative care terms occasionally do not translate well into other languages, and these translations may sound scary or stoke fear in patients or family members [7]. It is possible that the presentation of course material, and as a consequence practical implementation, may have more limitations if participants and/or local community members are less proficient in English in other LMICs.

The study is also subject to several sources of selection bias. First, this course was implemented at a single center at an urban, military teaching hospital in Accra, Ghana. It is possible that this population of students, and downstream the population of patients to whom these participants provide health care, are not generalizable to other patient populations, specifically those seen in other LMIC settings. It is also plausible that in another LMIC region, local cultural, religious, and health care practices differ significantly enough that implementation in these regions would not have the same effect and may lead to encountering unexpected boundaries. Second, the course was taken voluntarily, which may have led participants with a pre-existing interest in palliative care principles to self-select this course.

More importantly, participation in the one-year follow-up was also voluntary. Of the 33 course participants, only 11 individuals attended the one-year follow-up, and their responses and feedback may not be representative of the views or practices of the entire group. Additionally, this one-year follow-up group discussion was facilitated by the course director himself, which introduces further potential bias, specifically in the survey results and prompt responses elicited from course participants. In addition to these potential biases, the total number of participants in the class was low. It is worth noting that participation in this course was voluntary, and course participants were initially skeptical of the course material. After receiving positive feedback from participants, we expect interest in this course material to be significantly higher for future class enrollment. Providing this course material or that of a similar course to a larger number of participants in a future study would be highly valuable to the available literature.

Finally, this course, while initially designed for physicians, was delivered to a multidisciplinary group of participants, which may not be reflective of the health care practitioners who are best suited to use this material and incorporate it into their practice. However, it should be noted that in other LMIC settings, physicians are not necessarily the health care professionals who are in charge of breaking bad news and initiating palliative-care-focused discussions as a default [17]. At the facility at which participants are employed, it is commonplace for either clinical psychologists, nurses, or non-physician staff such as midwives or dieticians, to be heavily involved with end-of-life discussions, and non-physician participants cite that they are frequently the ones to break bad news about patient conditions. Participants noted that oftentimes, nursing or clinical psychology personnel have more time to have these conversations with families and that physicians, particularly the physicians who did not take the course, are often either too busy to have dedicated goals of care conversations or do not recognize the importance of such conversations. They also note that, since taking the course, they have felt more empowered to discuss with consulting physicians the need to have goals of care discussions and initiate palliative care-focused treatment.

## 5. Conclusions

Implementation of a curriculum with the intent to provide education on palliative medicine in LMICs is feasible and can result in local adoption of the practice of palliative care principles similar to those used in high-income settings. Our study suggests that the practice of these learned principles may enhance the relationships between patients, families, and health care professionals and may improve career satisfaction, specifically regarding the care provided at the end of life by health care professionals. Local religious and cultural idiosyncrasies were perceived as less of a barrier to the implementation of palliative care education and practice than previously thought. The widespread adoption of palliative care principles at the end of life in LMICs would benefit from further study: it may reduce health care-related costs, and prevent unnecessary hospitalizations, all while improving quality of life, perception of care at the end of life, as well as patient–clinician relationships.

## Figures and Tables

**Table 1 jcm-14-03643-t001:** Didactic curriculum overview. Days 1 and 2 of the didactic curriculum. These days were subdivided into two half-day sessions, each comprising three lectures and a relevant small group session. The small group session format included a case, with prompts to facilitate interdisciplinary discussion. Each small group included a course educator in the role of facilitator. * Indicates a small group session title. ^1^ TBI: traumatic brain injury. ^2^ EOL: end of life.

Day 1	Day 2
Principles of TBI ^1^: Part 1	Introduction to Palliative Care: Part 1
Principles of TBI ^1^: Part 2	Introduction to Palliative Care: Part 2
Therapies for TBI ^1^	Brain Death: Definitions and Exam
* TBI management	* Ethical, Religious and Cultural Issues in EOL ^2^ Care
Paroxysmal Sympathetic Hyperactivity	Shared Decision-Making
Persistent Coma and Delirium	Breaking Bad News
Neuroprognostication	Medications at End of Life
* Effective communication with families regarding prognosis	* Withdrawal of Life-Sustaining Therapy

**Table 2 jcm-14-03643-t002:** Paired cohort sample size and pre/post average test scores (complete pre/post data).

Group	N (%)	Pre-Test Average Score	Post-Test Average Score	Paired T-Test *p*-Value
Physician	7 (26%)	44.0	58.9	
Nurse	11 (41%)	31.6	49.1	
Psychologist	3 (11%)	30.7	45.3	
Other *	5 (19%)	37.6	45.6	
Missing	1 (4%)	28.0	24.0	
TOTAL	27 (100%)	35.7/100 (SD 9.8)	49.6/100 (SD 13.3)	<0.001

* other consists of dietician, midwife, and physician-clinical psychologist. See Appendix C for the full content of the pre-/post-test.

**Table 3 jcm-14-03643-t003:** Total cohort sample size and pre/post average survey scores.

Group	Pre N (%)	Post N (%)	Pre-Survey Average Question Score **	Post-Survey Average Question Score **
Physician	10 (31%)	8 (27%)	3.1	4.1
Nurse	12 (38%)	13 (43%)	2.8	3.7
Psychiatrist	4 (13%)	4 (13%)	2.4	3.8
Other *	5 (16%)	5 (17%)	2.2	3.7
Missing	1 (3%)	0 (0%)	1.5	-
TOTAL	32 (100%)	30 (100%)	2.7/5 (SD 0.9)	4.0/5 (SD 0.4)

* other consists of dietician, midwife, and physician-clinical psychologist. ** The pre-survey consisted of two questions while the post-survey consisted of nine questions (all using a scale of 1–5). See Appendix A and Appendix B for the full content of the pre-survey and post-survey.

**Table 4 jcm-14-03643-t004:** Comparison of pre-survey and one-year post-survey utilization of palliative care.

Question	Responses (N)	Average Score (Standard Deviation)	Median Score [Interquartile Range]
Pre-Survey	32	2.9 (1.2)	2.5 [2.0, 4.0]
One-Year Post-Survey	9	4.9 (0.3)	5.0 [5.0, 5.0]

Responses to pre-survey Q3 and one-year post-survey Q7, both of which prompt the respondents to the frequency of which they utilize principles of palliative care medicine in their medical practice. See Appendix A and Appendix B, and D for the full content of the pre-survey and post-survey questions.

## Data Availability

Data is unavailable to protect the privacy of participating health care providers, but is available upon request for the purpose of peer review.

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
