# Peer review of "End-of-Life Care Training for Patients with Traumatic Brain Injury in Ghana: A Novel Curriculum and Its Initial Implementation"

_jcm, 2025, doi:10.3390/jcm14113643_

Round 1
Reviewer 1 Report
Comments and Suggestions for Authors
This is an interesting study evaluating the importance of palliative care in Ghana especially LMICs. The scores of post-test were promising and valuable to the families of the patient and care givers. I have concerns regarding the study performed:
- How is this study different and unique than other studied published on palliative care in Ghana (other than just considering TBI patients)?
- Did authors noticed improved lifespan duration in patients receiving palliative care?
- Measures adopted to convince family members to start palliative care. What is the cost difference observed with and without palliative care. Highly recommend to plot a graph to show differences.
- Did authors observed halting or progression in overlapping of other medical conditions with palliative care?
- Compare intensity of care between curative and palliative care.
- Encourage to plot the data for line 156-166 in the graph format for easy visualization.
- Participants receiving palliative learning is low. Therefore, statistically data may not be that significant.
- Improving the length and duration of course and learning can promote better output. What are authors point of view?
Author Response
First and foremost, thank you for the detailed and highly thoughtful comments and suggestions.
We have attached a document with a point-by-point response to the feedback we have received.
We are looking forward to your response to our changes.
Sincerely,
John Bruno (on behalf of all co-authors)
- How is this study different and unique than other studied published on palliative care in Ghana (other than just considering TBI patients)?
While there are other studies previously published on palliative care in Ghana (and other SSA countries), to our knowledge, ours is the first publication on implementing an educational course focused on palliative care. We view this as a highly important aspect in which our publication differs from available literature, and we appreciate your feedback. We have added to the discussion to address these differences and provided more citations of available literature of palliative practice in Ghana to strengthen our claims.
- Did authors noticed improved lifespan duration in patients receiving palliative care?
One of the limitations of our study is that we did not record patient-centered outcomes related to our implementation of this palliative medicine course. For this reason, we did not collect data on the lifespan and/or mortality of patients receiving palliative care as a result of our course. We viewed this as beyond the scope of our initial intent with this publication, which was to explore the feasibility of such a course and the effect on health care providers. We have added a narrative of this limitation under our limitations section, and added citations of relevant previously available literature. We agree with the reviewer that patient-centered outcomes would be highly valuable, and we hope to explore this in future work we pursue on the topic.
3.Measures adopted to convince family members to start palliative care. What is the cost difference observed with and without palliative care. Highly recommend to plot a graph to show differences.
For the same reasons as item #2, we did not collect data on the cost of medical care for patients as a consequence of our course. We’ve added citations from previous literature, which would suggest this would be improved.
4. Did authors observed halting or progression in overlapping of other medical conditions with palliative care?
As mentioned in item 2, we did not obtain data on patient-centered outcomes in this study. However, we added this to our limitations
5.Compare intensity of care between curative and palliative care.
We appreciate this recommendation. We have added a line in the introduction addressing how palliative care can reduce patient exposure to unnecessary medicine of potentially higher intensity.
6.Encourage to plot the data for line 156-166 in the graph format for easy visualization.
Thank you for this recommendation. We have made some changes to this paragraph to make it easier to read. The full survey, along with each individual question’s grading scale, is available in the supplementary appendix, which we added a reference to in this paragraph.
7.Participants receiving palliative learning is low. Therefore, statistically data may not be that significant.
Thank you for your feedback. We agree that the overall class size was small, which is a limitation. While this class size is all that we were able to recruit for the initial course, considering the excellent feedback we have received from course participants, we expect enrollment to be higher in future courses we provide now that word has spread. We included a line in the limitations section addressing this low course participant enrollment
8.Improving the length and duration of course and learning can promote better output. What are authors point of view?
This is a highly important viewpoint. We, the authors, strongly agree with the reviewer on this statement. While we did not thoroughly divulge our future plans in the manuscript, we are actively revising, improving and expanding the course content for future courses based on feedback received on the initial implementation, and our assessment of palliative needs in other disease states (i.e. end stage cardiopulmonary failure, cancer, dementia, debilitating musculoskeletal disease) instead of predominantly TBI. We have high hopes that we will be able to reach a larger audience and cover a more diverse breadth of patient populations in our future endeavors.
Reviewer 2 Report
Comments and Suggestions for Authors
This manuscript describes the development and implementation of a novel palliative care and end-of-life (EOL) training curriculum for traumatic brain injury (TBI) in Ghana, along with promising results from pre/post-testing and a one-year follow-up. The study addresses a critical gap in low- and middle-income countries (LMICs), where palliative care education and access remain scarce. The interdisciplinary approach, focus on TBI—a high-burden condition in Ghana—and mixed-methods evaluation (quantitative scores, surveys, qualitative feedback) provide a robust foundation for the findings. The manuscript is well-structured, with clear objectives, methods, and conclusions that align with global efforts to advance palliative care as a human right.
Strengths
- Relevance and Timeliness: The research responds to urgent global health needs by addressing palliative care deficits in LMICs, particularly in the context of TBI, which is highly prevalent in Ghana due to motor vehicle crashes. The integration of cultural and religious considerations, often cited as barriers, adds nuance to discussions about feasible implementation in resource-limited settings.
- Curriculum Design and Multidisciplinary Engagement: The three-day curriculum, combining didactic lectures, small group sessions, and simulation, is thoughtfully tailored to LMIC constraints (e.g., omitting unavailable treatments). Involving nurses, psychologists, and physicians from diverse roles reflects real-world clinical teamwork, enhancing the training’s applicability. The SPIKES model integration for breaking bad news is a practical, evidence-based tool that participants reported using effectively.
- Robust Evaluation Methods: The pre/post-test score improvement (13.9-point increase, p < 0.001) demonstrates immediate knowledge gains, while the one-year follow-up provides valuable longitudinal data on sustained practice changes (e.g., frequent use of palliative communication strategies). Qualitative insights, such as improved job satisfaction and family interactions, highlight the curriculum’s impact beyond knowledge transfer.
- Policy and Practice Implications: The conclusion thoughtfully connects findings to broader goals, such as strengthening patient-provider relationships and challenging assumptions about cultural barriers. The emphasis on multidisciplinary training and local adaptation offers a replicable model for other LMICs.
Areas for Improvement
- Sample Size and Generalizability: While the study includes 33 participants, the one-year follow-up involves only 11 volunteers (33% of the cohort), raising concerns about selection bias. The urban military hospital setting may not represent rural or non-military healthcare contexts in Ghana or other LMICs. Discussing these limitations explicitly—perhaps suggesting future multicenter studies—would strengthen the manuscript.
- Cultural/Religious Nuance in Discussion: While the study notes that cultural/religious practices were less of a barrier than anticipated, the analysis could deepen by exploring specific examples of how training aligned with (or adapted to) Ghanaian norms. For instance, how did participants reconcile local beliefs with palliative care principles during EOL discussions? Including brief case vignettes from follow-up sessions would enrich this narrative.
- Shared Decision-Making Challenges: The underutilization of shared decision-making principles is an important finding, linked to financial constraints and family preferences. However, the discussion could expand on potential solutions, such as how to integrate evidence-based communication with local healthcare payment models (e.g., fee-for-service) or advocate for policy changes to improve access to affordable palliative medications.
- References and Compatibility: Ensure all references are correctly formatted (e.g., Table 3 lists “One-Year Post Survey Q7” but the methodology does not explicitly describe this question; clarify in Appendices). Additionally, cite recent LMIC-specific palliative care frameworks (e.g., WHO guidelines published after 2020) to enhance currency.
- Statistical Reporting: In Table 1, clarify the “Other” category (dieticians, midwives) and their role in TBI/EOL care, as their inclusion may influence interpretation of subgroup results (e.g., nurses scored lower pre-test but showed meaningful improvement). Report effect sizes (e.g., Cohen’s d) alongside p-values for a more comprehensive statistical picture.
This manuscript makes a valuable contribution to global health and palliative care literature, offering a pragmatic model for training in resource-limited settings. With minor revisions to address sample limitations, cultural nuance, and statistical clarity. The findings have direct relevance to researchers, educators, and policymakers working to advance EOL care in LMICs, and the curriculum design serves as a tangible example of evidence-based, locally adapted intervention.
Author Response
First and foremost, we want to thank you for the detailed and highly thoughtful comments and suggestions. We greatly appreciate the invitation to revise and resubmit our manuscript, and we believe that by incorporating the feedback we have received from both reviewers, our paper and its potential impact have become quite stronger.
In this letter, we respond point-by-point to all inquiries and comments. All changes made in the revised manuscript have been highlighted using the track changes feature.
We hope that our revised manuscript will now be suitable for publication in JCM.
We are looking forward to your response to our changes.
Sincerely,
John Bruno (on behalf of all co-authors)
*****************************************************
1. Sample Size and Generalizability: While the study includes 33 participants, the one-year follow-up involves only 11 volunteers (33% of the cohort), raising concerns about selection bias. The urban military hospital setting may not represent rural or non-military healthcare contexts in Ghana or other LMICs. Discussing these limitations explicitly—perhaps suggesting future multicenter studies—would strengthen the manuscript.
We agree and think it is important to mention that our low participant enrollment is a limitation that can be addressed in future trials. While this class size is all that we were able to recruit for the initial course, considering the excellent feedback we have received from course participants, we expect enrollment to be higher in future courses we provide now that word has spread. We have added several lines in the limitations section that address the need for future studies with larger participant enrollment
2. Cultural/Religious Nuance in Discussion: While the study notes that cultural/religious practices were less of a barrier than anticipated, the analysis could deepen by exploring specific examples of how training aligned with (or adapted to) Ghanaian norms. For instance, how did participants reconcile local beliefs with palliative care principles during EOL discussions? Including brief case vignettes from follow-up sessions would enrich this narrative.
Thank you for this recommendation. We agree that specific case/provider vignettes would have highly enhanced our discussion section. Unfortunately, while case vignettes were discussed at length during our one-year post-survey group meeting (with 11 participants), this session was not video recorded. We also never discussed with participants that we may include their specific vignettes. For these reasons, we decided not to include specific vignettes and provider statements in our discussion or results section.
3. Shared Decision-Making Challenges: The underutilization of shared decision-making principles is an important finding, linked to financial constraints and family preferences. However, the discussion could expand on potential solutions, such as how to integrate evidence-based communication with local healthcare payment models (e.g., fee-for-service) or advocate for policy changes to improve access to affordable palliative medications.
Thank you for this recommendation. We included a line under our discussion section addressing a potential role for shared decision making as palliative medicine provision evolves.
4. References and Compatibility: Ensure all references are correctly formatted (e.g., Table 3 lists “One-Year Post Survey Q7” but the methodology does not explicitly describe this question; clarify in Appendices). Additionally, cite recent LMIC-specific palliative care frameworks (e.g., WHO guidelines published after 2020) to enhance currency.
Thank you for this recommendation. With the many updates we made based on the excellent feedback we have received on our publications, we have added several new citations, including a 2021 update on palliative practice guidelines.
5. Statistical Reporting: In Table 1, clarify the “Other” category (dieticians, midwives) and their role in TBI/EOL care, as their inclusion may influence interpretation of subgroup results (e.g., nurses scored lower pre-test but showed meaningful improvement). Report effect sizes (e.g., Cohen’s d) alongside p-values for a more comprehensive statistical picture.
Thank you for these recommendations. We have added a line at the end of the discussion clarifying the “other” category and how they are involved in palliative care at Ghanaian hospitals.
Unfortunately, we do not have data available to calculate effect sizes alongside our P-values. We agree that it would be useful to collect this data for future studies on this topic, but we do believe that our data still adds value to the existing literature.
We additionally reconfigured the tables in regards to how physicians are separated, which we believe gives more power to our charts.
Round 2
Reviewer 1 Report
Comments and Suggestions for Authors
The manuscript in the current form is improved and can be considered for publication.